# Longitudinal Follow-Up of Blood Telomere Length in HIV-Exposed Uninfected Children Having Received One Year of Lopinavir/Ritonavir or Lamivudine as Prophylaxis

**DOI:** 10.3390/children8090796

**Published:** 2021-09-10

**Authors:** Audrey Monnin, Amélie Vizeneux, Nicolas Nagot, Sabrina Eymard-Duvernay, Nicolas Meda, Mandisa Singata-Madliki, Grace Ndeezi, James Kashugyera Tumwine, Chipepo Kankasa, Ameena Goga, Thorkild Tylleskär, Philippe Van de Perre, Jean-Pierre Molès

**Affiliations:** 1Pathogenèse et Contrôle des Infections Chroniques, INSERM U1058, Université Montpellier, Etablissement Français du Sang, University of Antilles, 34093 Montpellier, France; AUDREY-A91@hotmail.fr (A.M.); amelie.vizeneux@inserm.fr (A.V.); n-nagot@chu-montpellier.fr (N.N.); sabrina.eymard-duvernay@ird.fr (S.E.-D.); p-van_de_perre@chu-montpellier.fr (P.V.d.P.); 2Centre Muraz, Bobo-Dioulasso 01 P.O. Box 390, Burkina Faso; meda_nicolas@yahoo.fr; 3Effective Care Research Unit, Cecilia Makiwane Hospital, University of Fort Hare, East London 5207, South Africa; mandisa.singata@gmail.com; 4Department of Paediatrics and Child Health, School of Medicine, College of Health Sciences, Makerere University, Kampala P.O. Box 317, Uganda; gndeezi@gmail.com (G.N.); kabaleimc@gmail.com (J.K.T.); 5School of Medicine, Kabale University, Kabale P.O. Box 317, Uganda; 6Department of Paediatric and Child Health, University Teaching Hospital, University of Zambia School of Medicine, Lusaka P.O. Box 50110, Zambia; ckankasa@zamnet.zm; 7HIV Prevention Research Unit, South African Medical Research Council, Private Bag x385, Pretoria 0001, South Africa; Ameena.Goga@mrc.ac.za; 8Centre for International Health, Faculty of Medicine, University of Bergen, 5009 Bergen, Norway; thorkild.tylleskar@cih.uib.no

**Keywords:** HIV, children, prophylaxis, telomere shortening, growth, neuropsychological development, mitochondrial DNA, Africa

## Abstract

Telomere shortening can be enhanced upon human immunodeficiency virus (HIV) infection and by antiretroviral (ARV) exposures. The aim of this study was to evaluate the acute and long-term effect on telomere shortening of two ARV prophylaxes, lopinavir/ritonavir (LPV/r) and lamivudine (3TC), administered to children who are HIV-exposed uninfected (CHEU) to prevent HIV acquisition through breastfeeding during the first year of life, and to investigate the relationship between telomere shortening and health outcomes at six years of age. We included 198 CHEU and measured telomere length at seven days of life, at week-50 and at six years (year-6) using quantitative polymerase chain reaction. At week-50, telomere shortening was observed among 44.3% of CHEU, irrespective of the prophylactic treatment. Furthermore, this telomere shortening was neither associated with poor growth indicators nor neuropsychological outcomes at year-6, except for motor abilities (MABC test *n* = 127, β = −3.61, 95%CI: −7.08, −0.14; *p* = 0.04). Safety data on telomere shortening for infant HIV prophylaxis are scarce. Its association with reduced motor abilities deserves further attention among CHEU but also HIV-infected children receiving ARV treatment.

## 1. Introduction

Assaying the short- and long-term safety of antiretroviral (ARV) therapy is of utmost priority for public health guidelines on ARV regimens for HIV prevention and treatment. Given the urgent need to prevent and treat HIV, most ARV regimens have been established by a panel of experts using evidence from randomized, phase 3 clinical trials that have directly compared ARV regimens in research settings. Data on short-term safety of ARV therapy for pediatric use come from phase 1/2 safety and pharmacokinetic trials and nonrandomized, open-label studies [1]. Although pediatric ARV treatment regimens have proven effective in controlling HIV viral load and improving health outcomes in HIV-infected children, excess morbidities and mortalities are still reported in these pediatric populations [2,3,4]. The long-term side effects of ARV are poorly documented mainly due to the difficulty to truly discriminate drug effects from those of HIV exposure and/or infection.

Telomeres are the caretakers of our genetic information. These highly conserved DNA repeated sequences that cap the ends of the chromosomes prevent their degradation and their fusion, thus ensuring genomic stability [5]. Telomere shortening is a physiological event that occurs at each cell division in all somatic cells and is associated with age-related disorders [6]. Telomere dysfunction is the major cause of replicative senescence, but also initiates and maintains stem cell exhaustion and inflammation [7]. Many factors are associated with accelerated telomere shortening, such as male gender, smoking, alcohol consumption, ethnicity, obesity, sedentary lifestyle, genetic variants, socioeconomic status, as well as psychological stress [6,8]. Both ARV and HIV are known to promote telomere shortening. Chronic HIV-induced inflammation and free HIV viral particles themselves can enhance the production of reactive oxygen species damaging telomeric DNA and thus promoting telomere shortening [9,10,11]. Protease inhibitors (PI) [12,13,14,15,16], and particularly lopinavir [12], as well as nucleoside reverse transcriptase inhibitors induce reactive oxygen species production in several cell types or animal models [17,18,19,20]. A recent study reported a relationship between leucocyte telomere shortening and blood mitochondrial DNA content among HIV-infected or uninfected women, the greater the telomere shortening, the higher the mitochondrial DNA content [21].

If telomere shortening-associated premature aging was observed in adults living with HIV [22,23,24,25,26], relatively fewer studies investigated telomere length in younger populations. Of note, all but one of these studies were conducted outside Africa, the continent with the highest prevalence of HIV-infected children and children who are HIV-exposed uninfected (CHEU) [27]. One study reported shorter telomere length in early life for HIV-infected children when compared to children who are HIV-unexposed uninfected (CHUU), mostly of European origin [28], while another did not show a difference between HIV-infected and uninfected adolescents, mostly of black/African Canadian origin [29]. This discrepancy could possibly be explained by the age differences between cohorts and by the duration of ARV exposure. The impact of ARV exposure on telomere length was also addressed in CHEU in cross-sectional studies at different time points. Most of them described similar telomere length at birth between CHUU and CHEU who have been exposed in utero to zidovudine (AZT) [30], to the backbone AZT plus lamivudine (3TC) in combination with nevirapine (NVP), nelfinavir (NFV), or ritonavir-boosted protease inhibitor (PI) [31,32] or to other triple combination including abacavir (ABC), tenofovir disoproxyl fumarate (TDF) or emtricitabine (FTC) [31,32]. Later in childhood, at approximately two years of age, CHEU exposed in utero to AZT/3TC/NVP [28], AZT/3TC/PI [28,29], or TDF/FTC/PI [28] also presented similar telomere length when compared to CHUU. Some of these children also received short term (6-week) postnatal prophylaxis including AZT [28,29,30,32], AZT/3TC or AZT or ABC/FTC/NFV [29]. However, one study reported shorter telomere length in CHEU at six years of age that had been exposed in utero to 3TC plus lopinavir/ritonavir (LPV/r)-based regimens including ABC, AZT, or stavudine, compared to CHUU [33]. Only one longitudinal study showed telomere length shortening from birth to three years old among CHEU and CHUU [32]. More recently, ritonavir-boosted protease inhibitors were a risk factor for telomere shortening in HIV-infected pregnant women [34].

In a randomized controlled trial (PROMISE-PEP trial, NCT00640263), we tested the efficacy of single drug pre-exposure prophylaxis in infants to prevent mother-to-children transmission of HIV in breastfed CHEU [35,36]. The main outcomes of this study were percent HIV transmission and regimen safety. Furthermore, we conducted a follow-up study of these trial participants at six years of age to evaluate growth, clinical, and neurodevelopment outcomes post-ARV exposure (PROMISE-M&S trial, NCT03519503) [37]. These children received ARV prophylaxis of LPV/r or 3TC up to one year of age, providing a unique population to evaluate the impact of these drugs on telomere length among uninfected children, and without interference of other drugs from combined therapy.

The objectives of the present study were to compare the effects on telomere length of two ARV treatments, LPV/r and 3TC administered to prevent HIV acquisition in CHEU during the first year of life, and to assess whether telomere shortening at one year was predictive of impaired growth, clinical, and/or neurodevelopmental outcomes at six years of age.

## 2. Materials and Methods

### 2.1. Study Population

We performed a longitudinal observational study of CHEU enrolled in the PROMISE PEP trial (NCT00640263) and its follow-up study (PROMISE M&S, NCT03519503). The PROMISE PEP trial, conducted in Burkina Faso, South Africa, Uganda, and Zambia between November 2009 and May 2012, CHEU received daily LPV/r or 3TC prophylaxis to prevent mother-to-child-transmission of HIV-1 through breastfeeding [35,36]. These uninfected children at birth received seven days of NVP as per national guidelines and were thereafter randomly assigned to receive LPV/r or 3TC from seven days after birth (day-7) until one week after breastfeeding discontinuation, for a maximal duration of fifty weeks (W50). The PROMISE M&S study (recruitment from February 2017 to February 2018) consisted of a one- or two-day visit for a growth, clinical, and neuropsychological evaluation of the PROMISE PEP trial participants aged 5 to 7 years old who were HIV negative at the end of the prophylaxis period [37]. Among PROMISE M&S participants, we randomly selected 198 CHEU with 1:1 sex and prophylactic regimen ratios. The same selection criteria were used in a previous study investigating the mitochondrial DNA genotoxicity of the prophylaxis [38,39].

### 2.2. Sample Collection and DNA Extraction

Child dried blood spots (Whatman^®^903 cards, Lipomic Healthcare, New Delhi, India) were collected on study sites directly by heel prick for day-7 and week-50 time points (PROMISE PEP trial) or were processed from venous blood collected on EDTA tubes for the year-6 time point (PROMISE M&S trial). All dried blood spots were stored at −20 °C at the study sites in an individual zipped-pouch containing desiccant. DNA extraction was performed on 3-mm diameter punches (*n* = 3) using the QIAamp DNA Blood Mini Kit (Qiagen, Hilden, Germany) following the manufacturer’s instructions. Extracted DNAs were stored at −80 °C.

### 2.3. Telomere Length Assay

#### 2.3.1. Standard Curves and Quantitative Polymerase Chain Reaction

Telomere length measurement was performed by quantitative polymerase chain reaction, based on a method previously described by O’Callaghan and Fenech [40,41]. Two standard curves were used. The first curve was generated by a ten-fold serial dilution of a synthetic oligonucleotide of 14 repetitions of the telomeric sequence (TAAGGG) which determines the quantity of telomere in kilobases (kb) per reaction. The standard concentration ranged from 1.18 × 10^3^ to 1.18 × 10^7^ kb of telomere per reaction. The second curve consisted of a ten-fold serial dilution of a synthetic oligomer of the human single copy gene RPLP0 which estimates the number of copies of diploid genomes per reaction. The concentration ranged from 2.63 × 10^1^ to 2.63 × 10^5^ diploid genome copies. Oligomers and primers used for the quantitative polymerase chain reactions are described in the Appendix A.

Each quantitative polymerase chain reaction plate contained (i) both standard curves in duplicate, (ii) DNA (10 ng) from the Human Embryonic Kidney 293T cell line in triplicate to assess the intra- and inter-plate variations, and (iii) DNA (10 ng) extracted at day-7, week-50, and year-6 for each participant, in a single point assay for telomeric and RPLP0 regions. Amplification was performed on a LightCycler^®^480 II instrument (Roche, Basel, Switzerland) in a final volume of 30 µL containing 1× LightCycler^®^480 SYBR Green I Master (Roche, Basel, Switzerland) and 900 nM of both pairs of primers. The following thermal program was used: 95 °C for 10 min; 40 cycles of 95 °C for 15 s, 60 °C for 30 s, 72 °C for 30 s, and ended by a melting curve.

#### 2.3.2. Telomere Data Analysis

We performed absolute quantification analysis using the Fit Points method on a LightCycler^®^480 software (Roche, Basel, Switzerland). The threshold line was set at the beginning of the exponential phase of amplification. Telomere length per cell was calculated by dividing the Kb/reaction for telomere by the RPLP0 diploid copies/reaction. Results were expressed in Kb telomere per cell.

qPCR efficiency, intra- and inter-plate variation coefficients, quality control of quantitative polymerase chain reactions, and outlier identification are described in the Appendix A.

### 2.4. Statistical Analysis

We described the participant characteristics using either means with standard deviation (SD) or medians with interquartile range ([IQR]) for continuous variables, and percentages for categorical variables.

Our primary objective was to investigate the short-term and long-term effects of ARV prophylaxis on telomere length. We first described the telomere length at day-7, i.e., before the initiation of prophylactic regimens, and compared this parameter by study site and gender using the Kruskal–Wallis test applying the Dwass-Steel-Critchlow-Fligner method for all pairwise comparisons, and a Wilcoxon Mann–Whitney test, respectively.

Second, we addressed variation of telomere length between t0 (day-7 or week-50) and t1 (week-50 or year-6). We compared telomere length between the two time points using a Wilcoxon signed-rank test, and defined telomere shortening as follows:

Telomere shortening = [(TL at t1) + 4.03% × (TL at t1)] − [(TL at t0) − 4.03% × (TL at t0)] < 0, where 4.03% was the estimated experimental error (see Appendix A). This telomere shortening threshold closely approaches the physiological decline in telomere length reported by Ajaykumar et al. among Canadian CHEU [32].

We compared the proportion of CHEU with telomere shortening at week-50 (t0 = day-7) or at year-6 (t0 = week-50) between the two prophylactic regimens using a Chi-square test. Analyses for telomere shortening between week-50 and year-6 were restricted to three out of the four study sites (*n* = 128). The three sites were Burkina Faso, Uganda, and Zambia, because samples for South Africa were not collected at year-6.

Throughout the analyses, we compared the two prophylactic regimens, LPV/r and 3TC, using Student’s *t*-test or Wilcoxon Mann–Whitney test for continuous variables, and Chi-square test or Fisher’s exact test for categorical variables.

Our secondary objective was to assess whether telomere shortening at week-50 was associated with child health impairments at year-6 and with higher mitochondrial DNA content. For this purpose, we first investigated the association between telomere shortening at week-50 and growth indicators (weight-for-age Z-score, height-for-age Z-score and body mass index Z-score) as well as neuropsychological performances (global scores obtained from the Strengths and Difficulties Questionnaire, SDQ-25; the Test Of Variable of Attention, TOVA; the Movement Assessment Battery for Children second edition, MABC-2 and the Kaufman Assessment Battery for Children second edition, KABC-II) at year-6, using linear regressions. Methodology for scores analysis and main findings is described elsewhere [37]. A square transformation was applied to SDQ-25 score in order to obtain a normal distribution. We also investigated the association between hospital admissions since week-50 and telomere shortening at week-50 using Poisson regressions with robust error variance. Growth analyses were adjusted for the prophylaxis regimen, the age of the child, the gender, the gestational age, the study site, the type of income generating activities, the mother’s educational level, and the number of children under five years old living in the household. Neuropsychological and hospital admission analyses were adjusted for the same confounders plus the child’s weight and height at year-6. All of these analyses were restricted to Burkina Faso, Uganda, and Zambia for the above mentioned reason.

Second, we investigated the association between telomere length and shortening with mitochondrial DNA content using linear regression in an analysis restricted to CHEU from Burkina Faso and Uganda (*n* = 73). Zambia was excluded from the analyses because we previously reported an interaction between mitochondrial DNA content and platelet count [38,39]. Mitochondrial DNA depletion was defined as a 50% or more decrease in mitochondrial DNA copy number per cell from day-7 to week-50 [38]. Mitochondrial DNA content at day-7, week-50, and year-6 were incremented per 50 copies. Analyses at week-50 and year-6 were adjusted for the type of prophylactic regimen. To address whether telomere shortening at week-50 was associated with mitochondrial DNA depletion at week-50, we used log-binomial regressions adjusted for the type of prophylactic regimen.

Statistical analyses were performed using SAS studio (Copyright © 2021–2016, SAS Institute Inc., Cary, NC, USA). The forest plot was drawn using GraphPad software v7.0 (Copyright © 2021–2018).

## 3. Results

### 3.1. Characteristics of the Study Population

One hundred and sixty-seven CHEU were enrolled in our study after quality control of DNA (Appendix A). The characteristics of analyzed CHEU did not differ from those who were not selected, with the exception of lower platelet count (Appendix A). Children were equally distributed between the four sites of the trial (Table 1). Most of them were born at term. At day-7, mean height and weight were 49.6 ± 2.0 and 3.3 ± 0.5, respectively. However, 10.6% to 15.4% of CHEU had either stunted growth, were wasted, or underweight. Hemoglobin concentration and blood cell concentrations were within the normal range for more than 92.0% of CHEU. Although no difference between prophylactic groups was observed, the prevalence of CHEU with stunting or wasting was roughly two-fold higher for those who received LPV/r as compared to those who received 3TC. Characteristics of CHEU according to study sites are described in Appendix A.

According to the national prevention of mother-to-child transmission guidelines prevailing at the time the trial was conducted (2009–2012), none of the mothers had received an ARV treatment before the first antenatal visit. As per inclusion criteria, CD4 cell count was above 350 cells/mm^3^ (Table 2) and no mother was on ARV therapy. Zidovudine (ZDV) prophylaxis was the main ARV regimen given during pregnancy, followed by a combination ZDV plus 3TC. Viral load was controlled (less than 1000 copies/mL) in 59.3% of the mothers at day-7. Approximately 80.0% of mothers had an income-generating activity and 85.0% had attended school. Almost all mothers did not smoke during pregnancy or during the breastfeeding period (97.8 and 98.6%, respectively). However, more than one-third reported alcohol consumption during pregnancy. No maternal differences between prophylactic groups was observed at randomization. However, there seemed to be fewer mothers who had attended school seemed in the 3TC group as compared to those in the LPV/r group, and inversely for those who had a regular income-generating activity. Characteristics of the mothers according to study sites are described in Appendix A.

### 3.2. Telomere Length at Day-7

At randomization (day-7), median telomere length was 294 kb/cell with an interquartile range of 144 to 438 kb/cell (Table 3). Median telomere lengths (in kb/cell) for the 3TC and for the LPV/r groups were 270 [137; 434] and 321 [152; 445], respectively (*p* = 0.29). Telomere length was significantly different between sites (*p* < 0.01), CHEU from South Africa having the lowest telomere length compared to CHEU from the three other sites which were similar (*p* < 0.01 for South Africa versus Burkina Faso, South Africa versus Uganda and South Africa versus Zambia; *p* = 0.51 for Burkina Faso versus Uganda; *p* = 1.00 for Burkina Faso versus Zambia; *p* = 0.58 for Uganda versus Zambia). No statistical difference was observed in telomere length according to the 3TC and for the LPV/r groups at each site. No difference according to gender was observed (Appendix A).

### 3.3. Telomere Length after One Year of Prophylaxis

Characteristics of CHEU at week-50 are described in Appendix A. Overall, after one year of prophylaxis, telomere length remained stable as compared to baseline (day-7) values with a median of 294 kb/cell [141; 417] (*p* = 0.62) (Table 4). However, when dichotomizing telomere length by profile, 74 CHEU (44.3%) had telomere shortening at week-50, 34 (45.3%) were in the LPV/r group, and 40 (43.5%) were in the 3TC group (*p* = 0.81).

### 3.4. Telomere Length at Six Years Old

Characteristics of CHEU with available samples at year-6 (*n* = 128 CHEU from Burkina Faso, Uganda and Zambia) are described in Appendix A. No difference between prophylactic groups was observed. Among these children, telomere length decreased from week-50, for which the median was observed at 333 kb/cell [252; 444], to reach a median of 274 kb/cell [182; 368] at year-6 (*p* = 0.58). Eighty-six CHEU (67.2%) had telomere shortening at year-6, 43 in each prophylactic group (*p* = 0.61). Among these children, 25 (29.1%) had already demonstrated telomere shortening during the first year of life.

### 3.5. Health Outcomes at Six Years Old among CHEU with Telomere Shortening at Week-50

Characteristics at week-50 of CHEU with or without telomere shortening are described in Appendix A. No difference between the two populations was observed except for the gestational age (*p* < 0.01). Furthermore, the prevalence of prematurity seemed to be two-fold higher among CHEU without telomere shortening (*p* = 0.20) as compared to those with telomere shortening.

Linear and Poisson regression models with robust error variance after adjustment for confounders showed that telomere shortening at week-50 was neither associated with poor growth indicators at year-6 (Figure 1), nor hospital admissions since week-50 (PR = 0.75, 95%CI: 0.42, 1.35; *p* = 0.34). However, among neuropsychological performances, motor capacities assessed by the MABC-2 test were negatively associated with telomere shortening at week-50 (β = −3.61, 95%CI: −7.08, −0.14); *p* = 0.04) (Figure 1).

β values and their confidence intervals for the final global scores for the SDQ-25, TOVA, MABC-2, and KABC-II neuropsychological tests and are presented in Figure 1. A square transformation was applied to the SDQ-25 score. Growth analyses were adjusted for the prophylactic regimen, the age of the child, the gender, the study site, the type of income generating activities, the mother’s education level, and the number of children under five years old living in the household. Neuropsychological analyses were adjusted for the same cofounders plus the child’s height and weight at year-6. Abbreviations: WAZ, weight-for-age Z-score; HAZ, height-for-age Z-score; BMIZ, body mass index Z-score; SDQ-25, Strength and Difficulties Questionnaire; TOVA, Test of Variable Of Attention; MABC-2, Movement Assessment Battery for Children second edition; KABC-II, Kaufman Assessment Battery for Children second edition; CI, confidence interval.

### 3.6. Relationship between Telomere Length and Shortening with Mitochondrial DNA Content

We investigated the relationship between telomere length and mitochondrial DNA content in an analysis restricted to CHEU from Burkina Faso and Uganda (*n* = 73). At day-7, linear regression models showed that telomere length was not associated with mitochondrial DNA content (*n* = 73, β1 = 1.52, 95%CI: −9.28, 12.33; *p* = 0.78). At week-50 and after adjustment for the prophylactic regimen, we observed that the lower the mitochondrial DNA content, the higher the telomere length (*n* = 73, β1 = −7.2, 95%CI: −13.8, −0.72; *p* = 0.03), but at year-6 there was no significant association (*n* = 73, β1 = 2.95, 95%CI: −8.89, 14.78; *p* = 0.63). Finally, telomere shortening at week-50 was not associated with mitochondrial DNA depletion at week-50 after adjustment for the prophylactic regimen (*n* = 73, PR = 1.37, 95%CI: 0.86, 2.19; *p* = 0.19).

## 4. Discussion

In this study, CHEU receiving 3TC or LPV/r prophylaxis during breastfeeding exhibited similar telomere dynamics at one year of age. The prevalence of CHEU with telomere shortening at week-50 was similar between the two prophylactic regimens. Telomere length did not significantly decrease until the six-year benchmark. Finally, those identified with telomere shortening at week-50 did not demonstrate impaired growth or poorer clinical outcomes at year-6, but were associated with motor impairment.

Given the absence of a control group not receiving any prophylactic treatment in this study, these observations suggest that both of the study drugs could have the same effect on telomere length, or that the telomere attrition we report is physiological. Previous reports have failed to demonstrate ARV toxicity on telomere length among CHEU exposed in utero or for a short period post-partum (28–33), which could suggest that one year of 3TC or LPV/r prophylaxis does not modify the rate of telomere shortening. We also tentatively compared these observations with those that have already been published. It is noteworthy that there is an absence of consensus regarding a reference value for telomere length among different populations as well as within specific populations. Similarly, a consensus for the definition of telomere shortening is lacking. However, it is worth mentioning that there exists reassuring data in the use of short course, i.e., six weeks, infant ZDV prophylaxis on telomere length. Gianesin et al. showed similar telomere length between CHEU receiving prophylaxis compared to CHUU [28]. Furthermore, one longitudinal study did not show variation in telomere length from birth to 31 days among 58 Canadian CHEU receiving ZDV [32]. However, the latter study reported a quicker telomere length decrease during the first 40 weeks of life among 214 CHEU compared to CHUU [32]. Altogether, ARVs used in prophylaxis regimens are unlikely associated with telomere shortening.

We also took advantage of the cohort design of our study to evaluate the impact of telomere shortening early in life on child health and neurodevelopment. The definition of early biomarkers characterizing children with poor health outcomes is a major public health concern for CHEU [42]. A negative association between telomere shortening and motor capacity was observed. Reasons for such an association remain obscure but we believed it is worth repeating in different cohorts in order to more definitively conclude on its prognostic value.

This work is unique as it described telomere length evolution among an African cohort of CHEU with little interference with other maternal ARV exposure. Furthermore, given that the child prophylaxis consisted of a single drug, we were able to evaluate each drug individually and not in combination with other molecules. To date, only one other study has investigated telomere length among South African CHEU, using a cross-sectional design [33]. In addition, we followed CHEU to a later age than a previously published longitudinal study describing telomere length, which observed children until three years of age [32], and conducted original investigations on the relationship between telomere shortening and growth and neuropsychological development at school age. Furthermore, given that a small proportion of mothers has smoked during pregnancy and breastfeeding, our analysis obviates the major confounding factor which was commonly encountered in other studies among CHEU. Moreover, we benefited from the randomization of the PROMISE PEP trial thus minimizing the risk of selection bias, participant bias, and confounding factors bias. Telomere length measurement was well-performed according to a robust, standardized method with similar or lower inter- and intra-assay coefficients of variation as compared with the other studies reported so far among CHEU [27,29,31,32].

However, our study also presents limitations. First, in contrast to other studies carried out among CHEU or HIV-infected children that have assessed telomere length from whole venous blood, we measured telomere length from dried blood spots. A few studies support dried blood spots as a less invasive method for measuring this parameter but they report a high correlation between relative telomere length in dried bloods spots and whole venous blood [43,44]. Second, we have not been able to observe an effect of the gender on telomere length. Yet, the male gender is a well-documented factor associated with shorter telomere length in comparison to females in the general population [45,46,47]. However, it remains unclear whether this gender effect is already present at birth. Some studies have reported this observation in newborns [48,49] while others have not been able to find such a difference between male and female babies [50,51]. Among CHEU, one study reports an association between male gender and shorter telomere length at birth [32], which could not have been related to a sample size effect (*n* = 106 versus *n* = 167). However, one could speculate that this gender effect could be related to the ethnic disparity between the two studies, the first one encompassing CHEU from different origins (Indigenous, Black/African Canadian, White and Asian) while we enrolled only African CHEU. Ethnicity is a known factor associated with telomere length. Particularly, populations of African descent appear to have longer telomeres than Caucasian populations [52,53,54,55]. One study described similar telomere length between healthy black females and male newborns, as we observed in this study [54]. Larger studies are clearly warranted to further investigate both gender and ethnic differences concerning telomere length in newborns and children. Another limitation of our study is that, interestingly, telomere length at day-7 was around two-thirds lower for CHEU from South Africa as compared to those from the three other sites. To date, no study has reported such a difference due to the fact that none have been multi-centric. A recent study evaluated telomere length among seven Sub-Saharan populations (two ancestries from Botswana and Tanzania and three ancestries from Ethiopia) and showed that the San ancestry population from Botswana had a greater telomere length when compared to the six other populations. Third, we were unable to assess with accuracy at what point in time, after the first year of life, telomere shortening occurred as we did not have intermediate time points between the first and sixth year of life. Fourth, some information was self-reported, including previous clinical consultations and hospital admissions, smoking during pregnancy and breastfeeding, alcohol consumption during pregnancy, type of income, and the mother’s education. We also lack information on several factors which are known to be associated with telomere shortening such as paternal age [56,57,58], substance abuse [59,60,61], and cytomegalovirus infection [62]. Finally, we could not perform long-term follow-up of CHEU from South Africa because samples were not collected and stored.

In conclusion, our study revealed no significant difference regarding telomere length between the two prophylactic treatments, 3TC and LPV/r, after one year of treatment. However, the association between telomere shortening at W50 and motor impairment at year-6 deserves further attention for CHEU as well as for HIV-infected children.

## Figures and Tables

**Figure 1 children-08-00796-f001:**
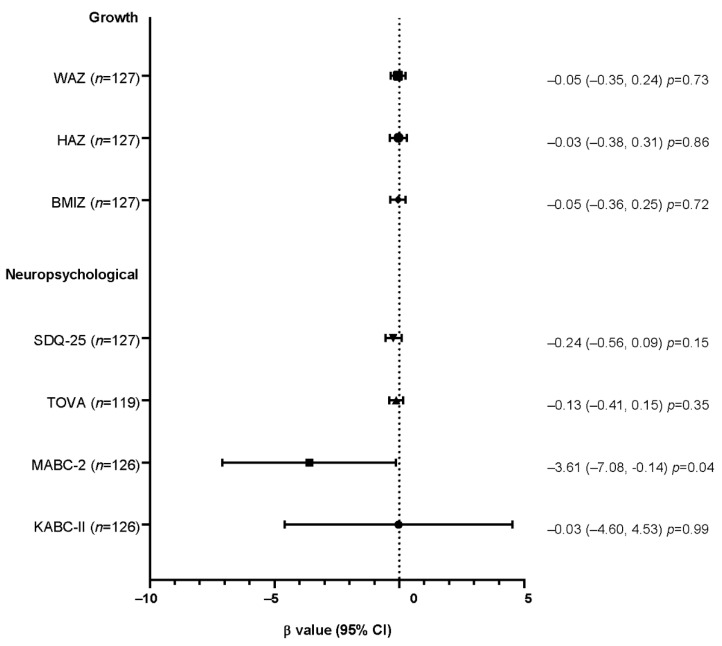
Forest plot of the association between growth and neuropsychological outcomes at year-6 with telomere shortening at week-50, assessed by linear regressions.

**Table 1 children-08-00796-t001:** Children’s characteristics at randomization (day-7).

Characteristics	LPV/r (*n* = 75)	3TC (*n* = 92)	Total (*n* = 167)	*p* Value ^a^
Socio-demographic				
Site; *n* (%)				0.40
Burkina Faso	21 (28.0)	23 (25.0)	44 (26.4)	
South Africa	13 (17.3)	26 (28.3)	39 (23.4)	
Uganda	20 (26.7)	23 (25.0)	43 (25.7)	
Zambia	21 (28.0)	20 (21.7)	41 (24.6)	
Gender; *n* (%)				0.59
Male	36 (48.0)	48 (52.2)	84 (50.3)	
Anthropometric				
Weight (kg); mean ± SD	3.2 ± 0.5 ^†^	3.3 ± 0.4	3.3 ± 0.5 ^†^	0.21
Height (cm); mean ± SD	49.4 ± 2.1	49.7 ± 1.9	49.6 ± 2.0	0.38
WAZ; mean ± SD	−0.8 ± 1.3 ^§^	−0.6 ± 1.2 ^†^	−0.7 ± 1.3 ^‡^	0.21
HAZ; mean ± SD	−1.0 ± 1.1 ^£^	−0.8 ± 1.1 ^£^	−0.9 ± 1.1 ^§^	0.32
WHZ; mean ± SD	−0.5 ± 1.4 ^§^	−0.3 ± 1.3 ^£^	−0.4 ± 1.3 ^ʊ^	0.40
Underweight (WAZ < −2); *n* (%)	13 (18.3) ^§^	12 (13.2) ^†^	25 (15.4) ^‡^	0.37
Stunting (HAZ < −2); *n* (%)	15 (20.6) ^£^	10 (11.1) ^£^	25 (15.3) ^§^	0.10
Wasting (WHZ < −2); *n* (%)	10 (14.1) ^§^	7 (7.8) ^£^	17 (10.6) ^ʊ^	0.20
Gestational age (week); median [IQR]	38.0 [38.0; 40.0]	38.0 [38.0; 40.0]	38.0 [38.0; 40.0]	0.37
Preterm birth (week); *n* (%)				0.47
No prematurity ≥ 37	65 (86.7)	83 (90.2)	148 (88.6)	
Prematurity < 37	10 (13.3)	9 (9.8)	19 (11.4)	
Hematological				
Hemoglobin (g/dL); mean ± SD	15.7 ± 2.1	15.8 ± 2.2 ^£^	15.8 ± 2.1 ^£^	0.82
Hemoglobin (g/dL); *n* (%)				0.52
Normal > 13	70 (93.3)	83 (91.2) ^†^	153 (92.2) ^†^	
Anemia ≤ 13	5 (6.7)	8 (8.8) ^†^	13 (7.8) ^†^	
Mild [12;13]	4 (5.3)	6 (6.6) ^†^	10 (6.0) ^†^	
Moderate [10;12]	-	2 (2.2) ^†^	2 (1.2) ^†^	
Very severe [0;9]	1 (1.3)	-	1 (0.6) ^†^	
Platelet count (10^3^/mm^3^); *n* (%)				0.18
Normal ≥ 125	68 (90.7)	84 (96.6) ^‡^	152 (93.8) ^‡^	
Thrombocytopenia < 125	7 (9.3)	3 (3.4) ^‡^	10 (6.2) ^‡^	
Mild [100;125]	5 (6.7)	2 (2.3) ^‡^	7 (4.3) ^‡^	
Moderate [50;100]	1 (1.3)	-	1 (0.6) ^‡^	
Severe [25;50]	-	1 (1.1) ^‡^	1 (0.6) ^‡^	
Very severe [0;25]	1 (1.3)	-	1 (0.6) ^‡^	
White cell count (10^3^/mm^3^); *n* (%)				NA
Normal > 2.5	75 (100.0)	91 (100.0) ^£^	166 (100.0) ^£^	
Neutrophil count (10^3^/mm^3^); *n* (%)				0.37
Normal > 1.5	71 (94.7)	87 (97.8) ^§^	158 (96.3) ^§^	
Neutropenia ≤ 1.5	4 (5.3)	2 (2.3) ^ƍ^	6 (3.6) ^ƍ^	
Mild [1.25;1.5]	2 (2.7)	2 (2.3) ^ƍ^	4 (2.4) ^ƍ^	
Moderate [1.0;1.25]	2 (2.7)	-	2 (1.2) ^ƍ^	

^†^ one missing value, ^§^ three missing values, ^‡^ five missing values, ^£^ two missing values, ^ʊ^ six missing values, ^ƍ^ three missing values. ^a^ Chi-square test or Fisher’s exact test as appropriate and Student’s *t*-test or Wilcoxon Mann-Whitney test for LPV/r versus 3TC. Abbreviations: SD, standard deviation; LPV/r, lopinavir/ritonavir; 3TC, lamivudine; IQR, interquartile range; WAZ, weight-for-age Z-score, HAZ, height-for-age Z-score; WHZ, weight-for-height Z-score, NA non-applicable.

**Table 2 children-08-00796-t002:** Maternal characteristics at randomization (day-7) and during the PROMISE PEP trial follow-up.

Characteristics	LPV/r (*n* = 75)	3TC (*n* = 92)	Total (*n* = 167)	*p* Value ^a^
At randomization (day-7)				
Socio-demographic characteristics				
Age (year); mean ± SD	29.6 ± 5.6	28.6 ± 5.4	29.0 ± 5.5	0.25
Parity; median [IQR]	3.0 [2.0; 4.0]	3.0 [1.0; 4.0]	3.0 [2.0; 4.0]	0.21
Children under 5 years living in the household; *n* (%)				0.02
0	41 (58.6) ^†^	37 (44.6) ^§^	78 (51.0) ^‡^	
1	17 (24.3) ^†^	39 (47.0) ^§^	56 (36.6) ^‡^	
2	10 (14.3) ^†^	6 (7.2) ^§^	16 (10.5) ^‡^	
3	2 (2.9) ^†^	1 (1.2) ^§^	3 (2.0) ^‡^	
Education				
Mother/caregiver ever attended school; *n* (%)				0.11
Yes	56 (80.0) ^†^	74 (89.2) ^§^	130 (85.0) ^‡^	
Economic status				
Type of income; *n* (%)				0.11
No income generating activities	15 (21.4) ^†^	13 (15.7) ^§^	28 (18.3) ^‡^	
Irregular income	36 (51.4) ^†^	34 (41.0) ^§^	70 (45.8) ^‡^	
Regular income	19 (27.1) ^†^	36 (43.4) ^§^	55 (34.0) ^‡^	
Clinical and biological characteristics				
BMI; median [IQR]	23.9 [21.7; 26.4]	23.9 [21.5; 28.0]	23.9 [21.4; 27.4]	0.51
CD4 cell count (cells/mm^3^); median [IQR]	490.0 [424.0; 601.0]	495.0 [430; 589.0]	494.0 [428.0; 595.0]	0.99
HIV viral load control; *n* (%)				0.52
<1000 copies/mL	42 (56.0)	57 (62.6) ^£^	99 (59.6) ^£^	
≥1000 copies/mL	33 (44.0)	34 (37.4) ^£^	67 (40.4) ^£^	
WHO HIV staging; *n* (%)				0.69
Stage 1	73 (97.3)	88 (95.6)	161 (96.4)	
Stage 2	2 (2.7)	4 (4.4)	6 (3.6)	
Maternal prophylaxis during pregnancy				
ARV regimen; *n* (%)				0.22
ZDV	65 (86.7)	79 (85.9)	144 (86.2)	
ZDV+3TC	5 (6.7)	11 (12.0)	16 (9.6)	
No ARV	5 (6.7)	2 (2.2)	7 (4.2)	
Duration of ARV prophylaxis taken during pregnancy (week); median [IQR]	10.0 [7.0; 12.0]	8.0 [5.0; 12.0]	9.0 [5.5; 12.0]	0.19
**During the PROMISE PEP trial**				
HIV viral load at week-38 (Log copies/mL); median [IQR]	4.0 [2.7; 5.0] ^§^	4.0 [2.2; 4.7] ^ʊ^	4.0 [2.5; 4.8] ^ƍ^	0.58
HIV viral load control at week-38; *n* (%)				0.62
<1000 copies/mL	20 (30.3) ^§^	24 (31.6) ^ʊ^	44 (31.0) ^ƍ^	
≥1000 copies/mL	46 (69.7) ^§^	52 (68.4) ^ʊ^	98 (69.0) ^ƍ^	
Duration of breastfeeding (week); median [IQR]	41.7 [33.7; 45.7]	43.4 [37.3; 47.9]	42.4 [35.9; 47.0]	0.08
Smoking during pregnancy; *n* (%)				1.00
No	62 (98.4) ^¤^	73 (97.3) ^¥^	135 (97.8) ^Ʃ^	
Smoking during breastfeeding; *n* (%)				1.00
No	62 (98.4) ^¤^	74 (98.7) ^¥^	136 (98.6) ^Ʃ^	
Alcohol consumption during pregnancy; *n* (%)				0.56
Yes	24 (38.1) ^¤^	25 (33.3) ^¥^	49 (35.5) ^Ʃ^	

^†^ five missing values, ^§^ nine missing values, ^‡^ fourteen missing values, ^£^ one missing value, ^ʊ^ eighteen missing values, ^ƍ^ twenty-five missing values, ^¤^ twelve missing values, ^¥^ seventeen missing values, ^Ʃ^ twenty-nine missing values. ^a^ Chi-square test or Fisher’s exact test as appropriate and Student’s *t*-test or Wilcoxon Mann–Whitney test for LPV/r versus 3TC. Abbreviations: LPV/r, lopinavir/ritonavir; 3TC, lamivudine; SD, standard deviation; IQR, interquartile range; BMI, body mass index; HIV, human immunodeficiency virus; WHO, World Health Organization; ARV, antiretroviral; ZDV, zidovudine.

**Table 3 children-08-00796-t003:** Children’ telomere length at day-7.

Site	PrEP	*n*	Telomere Length (kb/Cell) Median [IQR]	*p* Value ^a^
All	Any	167	294 [144; 438]	0.29
	LPV/r	75	321 [152; 445]	
	3TC	92	270 [137; 434]	
Burkina Faso	Any	44	335 [178; 463]	0.50
	LPV/r	21	293 [131; 424]	
	3TC	23	360 [231; 467]	
South Africa	Any	39	131 [112; 148]	0.65
	LPV/r	13	123 [79; 149]	
	3TC	26	132 [112; 147]	
Uganda	Any	43	391 [268; 485]	0.34
	LPV/r	20	420 [254; 529]	
	3TC	23	348 [268; 447]	
Zambia	Any	41	331 [268; 445]	0.63
	LPV/r	21	331 [295; 402]	
	3TC	20	321 [179; 461]	

Telomere measurement was performed on blood collected after randomization but before the administration of the first ARV prophylaxis dose. ^a^ Wilcoxon Mann–Whitney test for LPV/r versus 3TC. Abbreviations: PrEP, pre-exposure prophylaxis; kb, kilobase; IQR, interquartile range; LPV/r, lopinavir/ritonavir; 3TC, lamivudine.

**Table 4 children-08-00796-t004:** Telomere length at week-50 and proportion of CHEU with telomere shortening at week-50.

Site	PrEP	*n*	Telomere Length (kb/Cell) Median [IQR]	*p* Value ^a^	Children with Telomere Shortening *n* (%)	*p* Value ^b^
Burkina Faso	Any	44	333 [193; 463]	0.69	17 (38.6)	0.24
	LPV/r	21	292 [195; 483]		10 (47.6)	
	3TC	23	382 [191; 442]		7 (30.4)	
South Africa	Any	39	131 [108; 150]		17 (43.6)	
	LPV/r	13	125 [110; 152]	0.73	5 (38.5)	0.65
	3TC	26	137 [108; 148]		12 (46.2)	
	Any	43	332 [266; 431]		25 (58.1)	
Uganda	LPV/r	20	323 [292; 431]	0.91	12 (60.0)	0.82
	3TC	23	346 [266; 434]		13 (56.5)	
	Any	41	345 [266; 448]		15 (36.6)	
Zambia	LPV/r	21	345 [302; 433]	0.95	7 (33.3)	0.66
	3TC	20	338 [194; 498]		8 (40.0)	
All	Any	167	294 [141; 417]		74 (44.3)	
	LPV/r	75	305 [161; 430]	0.37	34 (45.3)	0.81
	3TC	92	267 [139; 417]		40 (43.5)	

^a^ Wilcoxon Mann–Whitney test for LPV/r versus 3TC. ^b^ Chi-square test for LPV/r versus 3TC. Abbreviations: CHEU, children who are HIV-exposed uninfected; PrEP, pre-exposure prophylaxis; kb, kilobase; IQR, interquartile range; LPV/r, lopinavir/ritonavir; 3TC, lamivudine.

## Data Availability

The full data set presented in this study is available on request from the corresponding author.

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
