# Peer review of "Longitudinal Follow-Up of Blood Telomere Length in HIV-Exposed Uninfected Children Having Received One Year of Lopinavir/Ritonavir or Lamivudine as Prophylaxis"

_children, 2021, doi:10.3390/children8090796_

Round 1

Reviewer 1 Report

This study is extremely important for people living with HIV, their children and for the definition of policies. Therefore, I would like to suggest that a larger and more diverse sample of as many countries as possible be included so that it can be more representative and that the selection be random. 

Author Response

Thank you for this comments. The population analysed herein represented about 25% of the recruited children in the PROMISE-PEP trial for Burkina Faso, South Africa and Uganda, but 10% of those of Zambia. We are not convinced that adding more children will strength the conclusion but as suggested by the reviewer, we will be pleased to collaborate with any other consortium to extend these conclusions to other countries, to children with other ARV exposure.

Reviewer 2 Report

Reviewer comments and suggestions

The current research paper investigates the acute and long-term effects of two antiretroviral exposure, lopinavir/ritonavir (LPV/r) and lamivudine (3TC), the drug was administered to children affected with HIV-exposed uninfected (CHEU) to prevent HIV acquisition through breastfeeding and to find out the relationship between telomere shortening and health outcomes at six years of age.

The result of the study noted telomere shortening was neither associated with poor growth indicators nor neuropsychological outcomes at year 6, except for motor abilities. Hence the study needs to thoroughly explore motor abilities in a long-term follow-up in relation to the shortening of telomere. 

Decision: Minor comments

Below are the comments for this paper to be incorporated in the revised version of the manuscript. 

  1. The first line of introduction ARV needs to be in full form “antiretroviral”
  2. Line 39-40 need a valid reference
  3. Line 64-65 The line need to be clearly written
  4. Line 73-74 more points needed to add here
  5. Line 88-89 The line needed to be present elsewhere as previous lines were related to 2 to 3year old kids
  6. Line 94-96 needed to add the outcome of the study
  7. Line 144 no need of references (38-39)
  8. Why the author thoroughly discussed the statistic, it was not needed. However, if they want they should mention the table or figure in the explained portion, that would be a benefit for readers
  9. Table 3 need to be modified in such a way that it can complete the information of study, very short title is not useful here
  10. Line 340 line need to be present some points related to the mention information
  11. Line 341-342 lines seems to be redundant
  12. Line 352-353 What the authors want to say here
  13. Line 378 Scarce is not a good choice with studies, please modify with a suitable one.
  14. Line 414 first time used please check the representation (TS)
  15. The references are not based on the MDPI journal, please check it again.

Author Response

We thank the reviewer to help to improve our manuscript.

  • The first line of introduction ARV needs to be in full form “antiretroviral”

The text was corrected accordingly.

  • Line 39-40 need a valid reference

A reference was added to the text.

  • Line 64-65 The line need to be clearly written

We modified the sentence as followed: “A recent study reported a relationship between leucocyte telomere shortening and blood mitochondrial DNA content among HIV-infected or uninfected women; the greater the telomere shortening, the higher the mitochondrial DNA content [20].

  • Line 73-74 more points needed to add here

We are not sure to understand this comment. We have added details that may help in understanding the message.

  • Line 88-89 The line needed to be present elsewhere as previous lines were related to 2 to 3year old kids

We agree with the reviewer. The sentence was placed earlier in the paragraph.

  • Line 94-96 needed to add the outcome of the study

The paragraph was restructured as followed: “In a randomized controlled trial (PROMISE-PEP trial, NCT00640263), we tested the efficacy of infant single drug pre-exposure prophylaxis to prevent mother-to-children transmission of HIV in breastfed CHEU [34,35]. The main outcomes of this study were the percentage of HIV transmission and the safety of the regimens. Furthermore, we conducted a follow-up study of these trial participants at the age of 6 years to evaluate growth, clinical and neurodevelopment outcomes post-ARV exposure (PROMISE-M&S trial, NCT03519503) [36]. These children received an ARV prophylaxis of LPV/r or 3TC up to one year of age, providing a unique population to evaluate the impact of these drugs on telomere length among uninfected children, and without interference with other drugs from combined therapy.”

  • Line 144 no need of references (38-39)

They have been removed.

  • Why the author thoroughly discussed the statistic, it was not needed. However, if they want they should mention the table or figure in the explained portion, that would be a benefit for readers

We agree with the reviewer that this section could be shortened. However, it allowed us to list all the variables and their defintion used herein and that were not described before. Also, because this work is an ancillary study using the PROMISE PEP and PROMISE M&S sample repository, some data points were missing for different reasons mentioned in the text which indeed have an impact in the analysis plan. Having these informations in the footnote of the tables or figures may be an option, however these are methods that we feel are more appropriate for this section.

  • Table 3 need to be modified in such a way that it can complete the information of study, very short title is not useful here

The text and table have been reorganized to maintain a logical reading flow.

  • Line 340 line need to be present some points related to the mention information

The sentence has been removed and replaced by “Because previous reports were not in favor of ARV toxicity on telomere length among CHEU exposed in utero or for a short period post-partum (28-33), one can suggest that one year of 3TC or LPV/r prophylaxis did not modify the rate of telomere shortening.”

  • Line 341-342 lines seems to be redundant

We agree with the reviewer the added information of the second sentence was rather subtle and have removed it.

  • Line 352-353 What the authors want to say here

The sentence was modified as followed: “Altogether, ARVs used in prophylaxis regimen are unlikely associated with accelerated telomere shortening”.

  • Line 378 Scarce is not a good choice with studies, please modify with a suitable one.

We replaced “Scarce” by “Few”.

  • Line 414 first time used please check the representation (TS)

We modified the text by using “telomere shortening”.

  • The references are not based on the MDPI journal, please check it again.

Thank you for this comment. We corrected the reference section according to Children MDPI format.